# PROMOTE: Prior-Guided Diffusion Model with Global-Local Contrastive Learning for Exemplar-Based Image Translation

## ABSTRACT

Exemplar-based image translation has garnered significant interest from researchers due to its broad applications in multimedia/multimodal processing. Existing methods primarily employ Euclidean-based losses to implicitly establish cross-domain correspondences between exemplar and conditional images, aiming to produce high-fidelity images. However, these methods often suffer from two challenges: 1) Insufficient excavation of domain-invariant features leads to low-quality cross-domain correspondences, and 2) Inaccurate correspondences result in errors propagated during the translation process due to a lack of reliable prior guidance. To tackle these issues, we propose a novel **pr**ior-guided diffusi**o**n **m**odel with gl**o**bal-local con**t**rastive l**e**arning (PROMOTE), which is trained in a self-supervised manner. Technically, global-local contrastive learning is designed to align two cross-domain images within hyperbolic space and reduce the gap between their semantic correlation distributions using the Fisher-Rao metric, allowing the visual encoders to extract domain-invariant features more effectively. Moreover, a prior-guided diffusion model is developed that propagates the structural prior to all timesteps in the diffusion process. It is optimized by a novel prior denoising loss, mathematically derived from the transitions modified by prior information in a self-supervised manner, successfully alleviating the impact of inaccurate correspondences on image translation. Extensive experiments conducted across seven datasets demonstrate that our proposed PROMOTE significantly exceeds state-of-the-art performance in diverse exemplar-based image translation tasks.

## CCS CONCEPTS

• **Computing methodologies → Artificial intelligence**; **Computer vision**; *Computer vision representations*;

## KEYWORDS

Prior, Diffusion Model, Contrastive Learning, Exemplar-based Imgae Translation

## 1 INTRODUCTION

Exemplar-based image translation aims to translate a user-provided conditional image, *e.g.* pose keypoints, edge maps, or strokes, into a realistic image with styles similar to those of an exemplar image [13, 43]. Compared to traditional image-to-image translation [12,

**Unpublished working draft. Not for distribution.**

44, 48], this task has increasingly attracted the attention of both academic and industrial communities due to its high controllability and flexibility [36]. Meanwhile, it finds applications in various fields including makeup transfer, social media, and the metaverse [24].

Early pioneering works [11, 30] primarily adopt generative adversarial networks (GANs) [6] for exemplar-based image translation, enabling global style control of the generated images. However, these methods ignore spatial correlations between a conditional image and an exemplar, which potentially leads to a lack of faithful details. Consequently, subsequent works [47] attempt to establish cross-domain correspondences between exemplars and conditional images to enhance local style control. They utilize contrastive losses [10, 42] or design adaptive networks [13], generating a warped exemplar to guide subsequent image synthesis by a generator. More recently, benefiting from the advancements of diffusion models for image generation, Seo et al. [36] first introduced the diffusion model to this task. This approach interleaves cross-domain matching with the diffusion step, iteratively refining a warped image to gradually reduce errors.

Overall, existing methods [36, 43, 47] for exemplar-based image translation generally follow a GAN/Diffusion-based matching-then-generation pipeline: two cross-domain images are first matched via a correspondence network to generate a warped image, and then a realistic image is translated by a generator optimized with several trivial Euclidean-based losses, as illustrated in Fig. 1 (a). Despite the remarkable success, these methods still suffer from two challenges: 1) *Insufficient excavation of domain-invariant features.* The Euclidean distance often fails to accurately measure the similarities between exemplars and conditional images due to the inherent domain gap. Relying solely on naive contrastive loss makes it difficult to effectively excavate domain-invariant features, potentially leading to sub-optimal correspondence. 2) *Lack of reliable prior guidance in the translation process.* Existing methods adopt progressive refinement that ignores reliable prior information from the target structure. Thus low-quality cross-domain correspondences will inevitably produce errors propagated throughout the generation step, significantly reducing the fidelity of the generated image.

Motivated by these issues, we propose a novel "**Pr**ior-guided Diffusi**o**n **M**odel with Gl**o**bal-Local Con**t**rastive L**e**arning" (PROMOTE) for exemplar-based image translation. The simplified framework of PROMOTE is depicted in Fig. 1 (b). In contrast to previous methods, our PROMOTE mainly designs global-local contrastive learning to effectively learn domain-invariant features within the hyperbolic space [18] and a prior-guided diffusion model that involves a self-supervised training scheme to generate realistic images. Specifically, to tackle the first challenge, we construct a hyperbolic space using exponential mapping [1] to facilitate the global alignment of representations between two cross-domain images. Compared to Euclidean space, hyperbolic geometry accounts for properties like curvature and nonlinearity, allowing it better to capture the

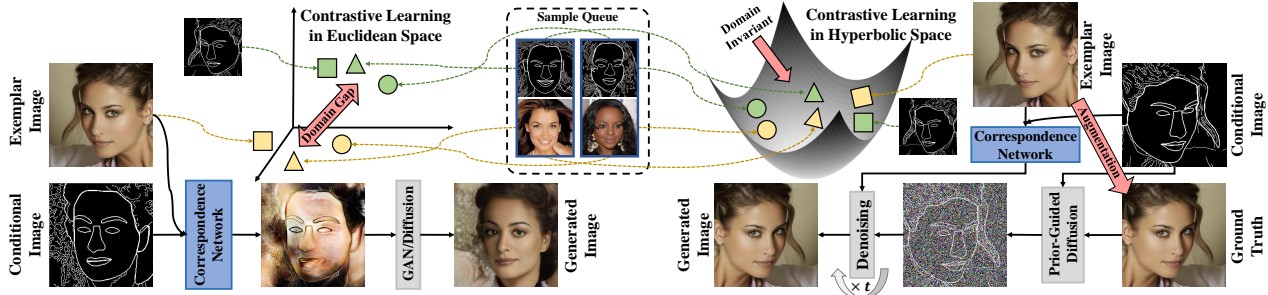

**Figure 1: Comparison of simplified frameworks between existing methods and our proposed PROMOTE. (a) Existing methods adopt GAN/Diffusion-based matching-then-generation pipelines optimized by Euclidean-based losses. (b) Our PROMOTE is a prior-guided diffusion model trained by reliable guidance that effectively extracts domain-invariant features through contrastive learning in hyperbolic space.**

consistent and discriminative features of cross-domain images and accurately measure their representation similarity. As an extension, a local contrastive loss defined by Fisher-Rao information [3, 31] is further designed to alleviate the domain gap by aligning their semantic correlation distributions, enabling exemplar and conditional encoders to more effectively capture domain-invariant features for subsequent translation. To address the second challenge, we leverage the structural information from the target, *i.e.* a conditional image, to modify the diffusion transition to generate prior noisy samples. We then mathematically derive a novel prior denoising loss to emphasize the preservation of the translated image's structure from the prior input. Notably, we employ several data augmentation strategies to construct ground truth and train our prior-guided diffusion model in a self-supervised manner, which results in more realistic images under reliable guidance. Extensive experiments conducted on seven datasets demonstrate that our PROMOTE achieves high-fidelity image translation and significantly outperforms state-of-the-art (SOTA) methods on all benchmarks. Our contributions are listed as follows:

- We propose global-local contrastive learning to effectively align two cross-domain images in hyperbolic space and reduce the representation gap between their semantic correlation distributions using the Fisher-Rao metric, thereby better extracting domain-invariant features.
- We propose a prior-guided diffusion model that leverages the structural prior from a target to modify the diffusion transitions. A novel prior denoising loss is mathematically derived by aligning the posterior and transition distributions, generating more realistic images.
- We are the first to construct reliable guidance in the diffusion model for this unsupervised task. Extensive experiments demonstrate that our PROMOTE significantly outperforms SOTA methods.

## 2 RELATED WORK

**Exemplar-based Image Translation.** Recently, exemplar-based image translation has attracted widespread attention [11, 26, 47]. Early works made use of global styles of exemplars to guide image generation. For instance, Park et al. [30] trained an encoder to transform an exemplar into a global style vector, which is then used to guide the image translation process. While this global control strategy ensures overall style coherence, it falls short in generating intricate details. Recently, numerous approaches [13, 24, 42] have arisen to build dense correspondences to control local detailed styles. Zhang et al. [43] established correspondences on a position-wise basis by using cosine attention to warp an exemplar accordingly, and then the warped image is fed to an image generation process in a manner of SPADE [30]. To explore more accurate correspondences, Zhan et al. [42] employed a marginal contrastive loss to explicitly explore domain-invariant features, and Jiang et al. [13] devised a masked adaptive transformer to suppress unreliable spatial matching and emphasize features of interest. Benefiting from the advance of diffusion models, Seo et al. [36] first introduced a diffusion model in this task, which interleaves the cross-domain matching and the diffusion step to iteratively refine warped images. Compared with these methods, we align cross-domain images in hyperbolic space instead of Euclidean space to learn better domain-invariant features. What's more, we implement image translation in a self-supervised manner by designing a prior-guided diffusion model, emphasizing the modification of diffusion and reverse transitions through reliable prior information from the target structure.

**Denoising Diffusion Probabilistic Model.** Arguing against the training instability and mode collapse of GANs [4, 11, 30], the denoising diffusion probabilistic model (DDPM) [9] generates high-quality samples by reversing the noising process. Notably, Nichol et al. [27] considered quality and speed, additionally predicting variance during the denoising process to improve the sampling process, while Song et al. [39] ensured fast and deterministic sampling by introducing a non-Markovian diffusion process. Rombach et al. [34] adopted an encoder-decoder network [7] to encode the input in a latent space and then trained the diffusion model to reduce computational complexity. Recently, some components containing external knowledge, *e.g.* classifiers [20, 37] and CLIP [19, 28], have been incorporated into the diffusion model to control the network for performance boosting. Besides revising network structure, Lee et al. [22] normalized the diffusion variables according to the timestep by leveraging fixed statistical information of conditional prior to improve the performance of speech synthesis. Differently, we exploit the target structure as instance-level prior to adaptively adjust

the mean and variance of diffusion distribution at each timestep. Besides, a novel prior denoising loss is mathematically derived to emphasize structure preservation in image translation.

## 3 METHOD

Given a conditional image $x_A$ from domain $\mathcal{A}$ and an exemplar image $y_B$ from domain $\mathcal{B}$, the exemplar-based image translation task aims to generate a target image $x_B$ that preserves the semantic structure of $x_A$ but adopts the style of $y_B$. The overall architecture of our proposed PROMOTE is illustrated in Fig. 2, which mainly consists of two components: the visual encoders with global-local contrastive learning for extracting robust domain-invariant features and a prior-guided diffusion model for producing high-fidelity images. Notably, the positive image $y_A$ from domain $\mathcal{A}$ is derived from $y_B$ for global-local contrastive learning. Unlike previous methods where $x_A$ and $x_B$ are irrelevant with $y_A$ and $y_B$, our pipeline applies the same data augmentation techniques to $y_A$ and $y_B$, generating credible $x_A$ and $x_B$ to train our diffusion model in a self-supervised manner.

### 3.1 Global-Local Contrastive Learning for Robust Cross-domain Correspondence

Given $x_A$ and $y_B$, the visual domain-invariant representations $\mathbf{X}_A \in \mathbb{R}^{C \times H \times W}$ and $\mathbf{Y}_B \in \mathbb{R}^{C \times H \times W}$ are extracted by the conditional and exemplar encoders $\varepsilon_A$ and $\varepsilon_B$, respectively, with $C$ dimensions, height $H$ and width $W$ following [13]. To effectively build the cross-domain correspondence between two images, $\mathbf{X}_A$ and $\mathbf{Y}_B$ are expected to contain accurate and sufficient domain-invariant features [42]. Previous methods mainly apply Euclidean-based contrastive loss [13, 42] to minimize the global similarity between $\mathbf{Y}_B$ and its structurally identical conjugate representation $\mathbf{Y}_A$ to alleviate the domain shift. However, Euclidean space is very sensitive to the overall content and semantics of images, which means that even if two images with different styles have similar structures, their Euclidean distance is still very large, resulting in ineffective representations of images from different domains. Therefore, we propose a novel global-local contrastive learning framework for explicitly excavating domain-invariant features.

**Global Contrastive Learning in Hyperbolic Geometry.** Inspired by the success of hyperbolic embedding in image retrieval [1, 18], we motivate learning the contrastive and discriminant properties of cross-domain images in hyperbolic geometry. Unlike Euclidean space $\mathbb{R}^n$, hyperbolic space $\mathbb{D}^n$, as a Riemannian manifold [41] with constant negative curvature, derives some geometric properties such as curvature, symmetry, and nonlinearity [18], which can better capture the domain-invariant features and improve the consistency of cross-domain visual representations [14]. Given a Euclidean vector $\mathbf{v}$, the exponential bijective mapping [18] $\exp_p : T_p \mathbb{R}^n \to \mathbb{D}^n$ of the Poincaré ball model [29] $\mathbb{D}^n := \{\mathbf{v} \in \mathbb{R}^n \mid c\|\mathbf{v}\|^2 < 1, c \geq 1\}$ with the curvature $c$ projects $\mathbf{v}$ into hyperbolic space, denoted as:

$$\exp_p(\mathbf{v}) := p \oplus \left( \tanh\left( \sqrt{c} \frac{\lambda_p \|\mathbf{v}\|}{2} \right) \frac{\mathbf{v}}{\sqrt{c}\|\mathbf{v}\|} \right), \quad (1)$$

where $\oplus$ is the differentiable Möbius addition, $p \in \mathbb{D}^n$ is the reference point empirically set to 0 for simplified computation, and

$\lambda_p = \frac{2}{1-c\|\mathbf{v}\|}^2$ denotes the conformal factor that scales the local distance. Specifically, we follow Eq. (1) to obtain the hyperbolic representations $\overline{\mathbf{Y}}_A$ and $\overline{\mathbf{Y}}_B$ by applying the exponential mapping to $\mathbf{Y}_A$ and $\mathbf{Y}_B$ before performing the contrastive operation as illustrated in Fig. 2.

The hyperbolic geometry is globally differential to the Euclidean and provides the closed-form distance equation between two points in the hyperbolic space [5]:

$$d_{\mathrm{hs}}(a, b) = \frac{2}{\sqrt{c}} \arctan(\sqrt{c}\| - a \oplus b\|). \quad (2)$$

Then we replace cosine similarity [8, 33] with the distance of Eq. (2) to calculate the global contrastive loss of the cross-domain representations $\overline{\mathbf{Y}}_A$, $\overline{\mathbf{Y}}_B$ in the hyperbolic space:

$$\mathcal{L}_{\mathrm{global}} = -\log \frac{\exp(d_{\mathrm{hs}}(\overline{\mathbf{Y}}_A, \overline{\mathbf{Y}}_B))}{\exp(d_{\mathrm{hs}}(\overline{\mathbf{Y}}_A, \overline{\mathbf{Y}}_B)) + \sum_n \exp(d_{\mathrm{hs}}(\overline{\mathbf{Y}_n}, \overline{\mathbf{Y}}_B))}, \quad (3)$$

where $\overline{\mathbf{Y}_n}$ $(n = A, B)$ are randomly selected negative samples from the same training batch, half of which come from domain $\mathcal{A}$ and half from domain $\mathcal{B}$.

**Patch-level Contrastive Learning with Semantic Correlation Alignment.** Aligning cross-domain images solely at the image level is insufficient as it fails to address the challenges associated with exploring the semantic correlation of appearance attributes, which are crucial for exemplar-based image translation. Thus we design a novel patch-level contrastive learning approach to capture domain-invariant features in a fine-grained manner, as depicted in Fig. 3. Intuitively, semantic correlations among attributes (*e.g.* the relative positions of the eyes, nose, and mouth) within an image can be described by the similarity distribution [15]. Given two cross-domain images $y_A$ and $y_B$ with consistent structures, we propose to bridge their semantic correlation distributions via the shortest path in probability space, which is expected to be sufficiently small.

Practically, we first randomly select $K$ patches in $y_A$ and calculate the semantic correlation distribution $\mathbf{D}_A^k$ for the $k$-th patch $y_A^k$ in relation to other patches within the hyperbolic space:

$$\mathbf{D}_A^k = [d_{\mathrm{hs}}(\overline{\mathbf{Y}}_A^k, \overline{\mathbf{Y}}_A^1), d_{\mathrm{hs}}(\overline{\mathbf{Y}}_A^k, \overline{\mathbf{Y}}_A^2), ..., d_{\mathrm{hs}}(\overline{\mathbf{Y}}_A^k, \overline{\mathbf{Y}}_A^K)]^{\mathrm{T}}, \quad (4)$$

Similarly, we extract $K$ patches at the same location of $y_B$ and obtain its similarity distribution $\mathbf{D}_B^k$:

$$\mathbf{D}_B^k = [d_{\mathrm{hs}}(\overline{\mathbf{Y}}_B^k, \overline{\mathbf{Y}}_B^1), d_{\mathrm{hs}}(\overline{\mathbf{Y}}_B^k, \overline{\mathbf{Y}}_A^2), ..., d_{\mathrm{hs}}(\overline{\mathbf{Y}}_B^k, \overline{\mathbf{Y}}_B^K)]^{\mathrm{T}}. \quad (5)$$

To align the semantic correlation distributions between the two images from different domains, we impose the consistency constrain on $\mathbf{D}_A^k$ and $\mathbf{D}_B^k$ for all $K$ sampled patches by utilizing the shortest path, which can be given in closed-form $p_{\mathrm{FR}}$ under the Fisher-Rao information metric and Gaussian assumption, as illustrated in Theorem 3.1:

$$\mathcal{L}_{\mathrm{local}} = \sum_{k=1}^{K} p_{\mathrm{FR}}(\mathbf{D}_A^k, \mathbf{D}_B^k). \quad (6)$$

THEOREM 3.1. *Consider two independent Gaussian distributions: $g_1$ with mean $\mu_1$ and standard deviation $\sigma_1$, and $g_2$ with mean $\mu_2$ and standard deviation $\sigma_2$. Given the Riemannian metric defined by the Fisher-Rao information [3], the closed-form expression for the shortest*

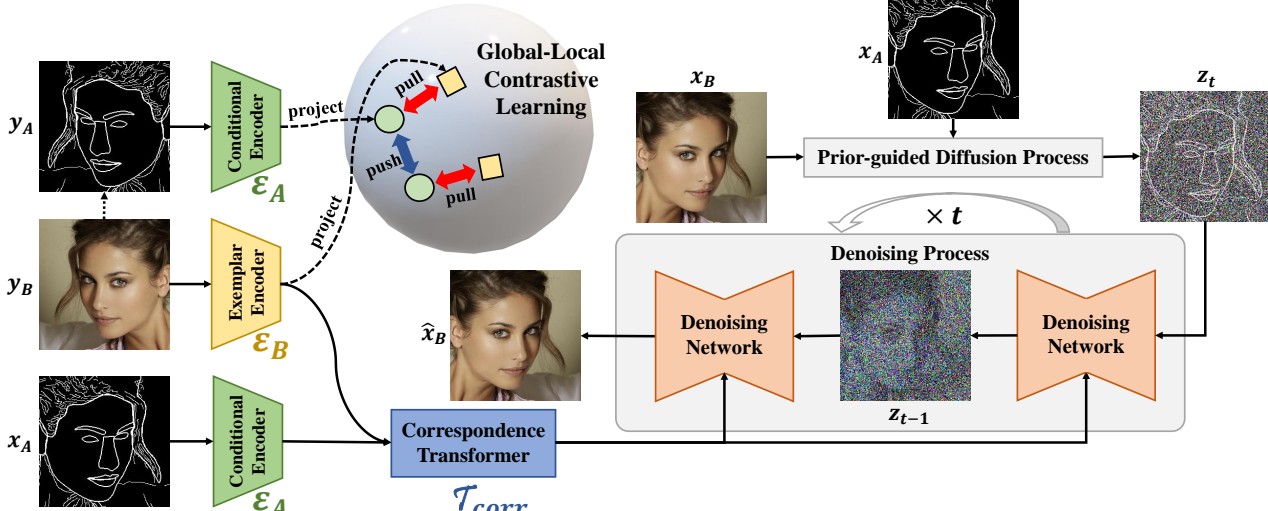

Figure 2: The framework of our proposed method consists of conditional and exemplar encoders with global-local contrastive learning, a correspondence transformer, and a self-supervised prior-guided diffusion model. For encoders, $x_A$, $y_B$, and $y_A$ denote the conditional image, the exemplar image, and the positive image, respectively. For the diffusion model, $x_B$, $z_t$, and $\hat{x}_B$ denote the self-supervised ground truth, the prior-guided noise sample at timestep $t$ and the denoising sample, respectively.

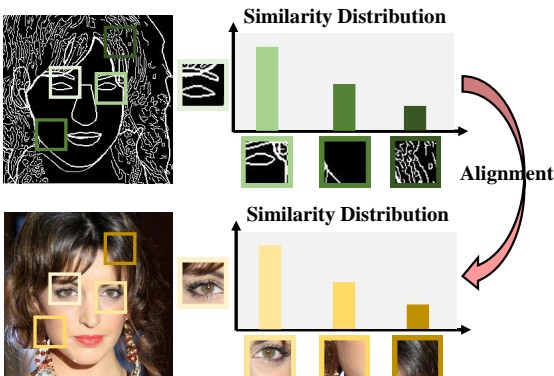

Figure 3: The illustration of semantic correlation alignment in our proposed patch-level contrastive learning.

*path between these two Gaussian distributions is given by:*

$$p_{FR}(g_1, g_2) = 2\sqrt{2}\arctan\left(\sqrt{\frac{(\mu_1 - \mu_2)^2 + 2(\sigma_1 - \sigma_2)^2}{(\mu_1 - \mu_2)^2 + 2(\sigma_1 + \sigma_2)^2}}\right). \quad (7)$$

The proof is provided in Supplementary Material A. Notably, a balancing weight $\gamma$ is employed to control the global and local alignment, and the total contrastive loss $\mathcal{L}_{cl}$ is:

$$\mathcal{L}_{cl} = \mathcal{L}_{global} + \gamma\mathcal{L}_{local}. \quad (8)$$

Benefiting from the global-local contrastive learning in the hyperbolic feature space, visual encoders can more accurately excavate domain-invariant features. Then, $\mathbf{X}_A$ is mapped to query $\mathbf{Q} \in \mathbb{R}^{HW \times C}$, $\mathbf{Y}_B$ is mapped to key $\mathbf{K} \in \mathbb{R}^{HW \times C}$ and value $\mathbf{V} \in \mathbb{R}^{HW \times C}$. These are fed into the correspondence transformer $\mathcal{T}_{corr}$, which produces the warping estimation $\widetilde{\mathbf{A}} \in \mathbb{R}^{HW \times HW}$ and the

correspondence feature map $\mathbf{X}_{corr} \in \mathbb{R}^{HW \times C}$ following [13]. Since our approach effectively bridges the domain gap between the domain $A$ and $B$ to extract domain-invariant features of $\mathbf{X}_A$ and $\mathbf{Y}_B$, it facilitates the accurate feature mapping by the correspondence transformer, thereby providing enhanced visual features for the subsequent image reconstruction.

## 3.2 Prior-guided Diffusion Model

In this section, we propose a novel prior-guided diffusion model trained in a self-supervised manner for exemplar-based image translation.

**Self-supervised Training.** Due to the lack of ground truth as a reference in exemplar-based image translation, most previous methods employ the learned correspondences to warp exemplars and implicitly constrain the structure and content of the generated images using several trivial losses. Inspired by [40], we train our diffusion model in a self-supervised manner by applying several data augmentation techniques (including horizontal flipping, rotation, and elastic deformation) to $y_B$ and $y_A$. This generates a ground truth sample $x_B$ and its conjugate image $x_A$, which are incorporated into the training scheme. Benefiting from the deterministic guidance of ground truth, the adverse impact of low-quality cross-domain correspondence during image translation is alleviated, which could avoid reliance on several trivial losses typically used by previous methods. Besides, we believe that this self-supervised training compels the network to comprehend the complex nonlinear mapping between cross-domain correspondence $\mathbf{X}_{corr}$ and translated image $\hat{x}_B$, thereby producing high-fidelity images.

**Prior-guided Forward Process.** Aiming to produce an image $\hat{x}_B$ that matches the style of $y_B$ and maintains the structure of $x_A$, the prior-guided forward process leverages $x_A$ to modify the

forward prior and generates a new Markov diffusion chain $\tilde{q}$ to corrode $x_B$ (denoted as $z_{0:T}$ for brevity) into a non-standard Gaussian distribution $\mathcal{N}(r, \eta^2 \mathbf{I})$:

$$\tilde{q}(z_t | z_{t-1}) := \mathcal{N}(z_t; \sqrt{\alpha_t} z_{t-1} + (1 - \sqrt{\alpha_t}) r, \beta_t \eta^2 \mathbf{I}), \quad (9)$$

where $\{\beta_t\}_{t=1}^T$ denotes the variance schedule provided in DDPM [9], $\alpha_t = 1 - \beta_t$, $r = (1-\eta) x_A$, and $\eta$ is coefficient schedule that controls the prior intensity. Compared with regular Markov transitions, Eq. (9) can adaptively adjust the mean and variance of the distribution of diffusion target $z_t$ at each timestep $t$ based on the structural prior $r$. We also admit sampling $z_t$ given by $z_0$ for any timestep $t$ in closed-form:

$$\tilde{q}(z_t | z_0) = \mathcal{N}(z_t; \sqrt{\bar{\alpha}_t} z_0 + \sqrt{1 - \bar{\alpha}_t} r, (1 - \bar{\alpha}_t) \eta^2 \mathbf{I}). \quad (10)$$

where $\bar{\alpha}_t = \prod_{i=1}^t \alpha_i$. At timestep $T$, $z_T = \sqrt{\bar{\alpha}_T} z_0 + \sqrt{1 - \bar{\alpha}_T} \eta \epsilon + \sqrt{1 - \bar{\alpha}_T} r = \eta \epsilon + r \sim \mathcal{N}(r, \eta^2 \mathbf{I})$, where $\epsilon \sim \mathcal{N}(0, \mathbf{I})$. Unlike conventional diffusion trajectories, the diffusion distribution at the end step evolves into a non-standard Gaussian distribution $\mathcal{N}(r, \eta^2 \mathbf{I})$. Notably, we set $\eta = \sqrt{\bar{\alpha}_t}(1 - \sqrt{\bar{\alpha}_t})$ as a quadratic schedule to encourage random noise to corrupt the overall style and texture in early phases of the diffusion process, and then emphasize the structural prior in later steps, allowing our diffusion model to effectively preserve target semantic information during the generation process.

**Denoising Loss for Reverse Process.** The reverse process of the diffusion model essentially involves aligning the means of posterior and transition distributions [2]. Therefore, we combine Eq. (9) and Eq. (10) to derive a tractable posterior (see Supplementary Material B for detailed derivation):

$$\tilde{q}(z_{t-1} | z_t, z_0) = \frac{\tilde{q}(z_t | z_{t-1}, z_0) \cdot \tilde{q}(z_{t-1} | z_0)}{\tilde{q}(z_t | z_0)},$$

$$:= \mathcal{N}(z_{t-1}; \tilde{\mu}_q(z_t, z_0), \tilde{\Sigma}_q(z_t, z_0)),$$

$$\text{where } \tilde{\mu}_q(z_t, z_0) = \frac{\sqrt{\alpha_t} \delta_{t-1} z_t + \sqrt{\bar{\alpha}_{t-1}} \beta_t z_0}{\delta_t} \quad (11)$$

$$+ \frac{2\delta_{t-1} \beta_t - \sqrt{\alpha_t} \delta_{t-1} \eta}{\delta_t} r,$$

$$\delta_t = 1 - \bar{\alpha}_t, \text{ and } \tilde{\Sigma}_q(z_t, z_0) = \frac{\delta_{t-1} \beta_t}{\delta_t} \eta^2.$$

We next parameterize $\tilde{p}_\theta(z_{t-1} | z_t) := \mathcal{N}(z_{t-1}; \tilde{\mu}_\theta(z_t), \tilde{\Sigma}_\theta(z_t) \mathbf{I})$ by employing a network and leverage the KL-divergence to minimize the difference between $\tilde{p}_\theta(z_{t-1} | z_t)$ and the posterior distribution $\tilde{q}(z_{t-1} | z_t, z_0)$. Since $\tilde{\Sigma}_q(z_t, z_0)$ and $\tilde{\Sigma}_\theta(z_t)$ are variable-irrelevant constant terms, by aligning $\tilde{\mu}_q(z_t, z_0)$ and $\tilde{\mu}_\theta(z_t)$, we can derive the prior denoising loss $\mathcal{L}_{\text{diff}}$ of our diffusion model (see Supplementary Material C):

$$\mathcal{L}_{\text{diff}} = \|\tilde{\mu}_q(z_t, z_0) - \tilde{\mu}_\theta(z_t)\|^2$$

$$= \|\rho_t r + \epsilon - \epsilon_\theta(\sqrt{\bar{\alpha}_t} z_0 + \sqrt{1 - \bar{\alpha}_t} \eta \epsilon + \sqrt{1 - \bar{\alpha}_t} r, t)\|^2, \quad (12)$$

where $\rho_t = \frac{2\delta_{t-1} \beta_t - \sqrt{\alpha_t} \delta_{t-1} \eta}{\delta_t}$. The first two terms of our denoising loss can be viewed as adaptive modifications to the noises during the diffusion process based on prior, forcing our model to emphasize the target structure and improve the sampling quality under the prior guidance.

## 3.3 Loss Functions

In addition to the diffusion loss for denoising and the global-local contrastive loss for learning domain-invariant features, two necessary loss functions are incorporated to train our network in an end-to-end way following [24].

**Correspondence Loss.** Intuitively, we use the reliable correspondence $\widetilde{\mathbf{A}}$ extracted by the correspondence transformer $\mathcal{T}_{\text{corr}}$ to warp the exemplar $y_B$, expecting the resulting image should be as close to the ground truth $x_B$ as possible. Therefore, the optimization objective for the learned correspondence is denoted as:

$$\mathcal{L}_{\text{corr}} = \|\widetilde{\mathbf{A}}^T \cdot \text{DS}(y_B) - \text{DS}(x_B)\|_1, \quad (13)$$

where $\text{DS}(\cdot)$ denotes the down-sampling operation used to resize the images $y_B$ and $x_B$.

**Cycle-Consistency Loss.** To ensure that visual information is not discarded during the warping process, the warped exemplar is expected to be recovered to the original exemplar $y_B$ under the guidance of the inverse correspondence, defined as the cycle-consistency loss [48]:

$$\mathcal{L}_{\text{cyc}} = \|\widetilde{\mathbf{A}} \cdot \widetilde{\mathbf{A}}^T \cdot \text{DS}(y_B) - \text{DS}(y_B)\|_1. \quad (14)$$

**Total Loss.** Benefiting from the self-supervised training, our approach constructs ground truth to explicitly guide the image translation, compared to unsupervised schemes, which could produce more faithful images. The overall optimization objective is expressed as:

$$\mathcal{L}_{\text{total}} = \mathcal{L}_{\text{diff}} + \lambda_1 \mathcal{L}_{\text{cl}} + \lambda_2 \mathcal{L}_{\text{corr}} + \lambda_3 \mathcal{L}_{\text{cyc}}, \quad (15)$$

where $\lambda_1$, $\lambda_2$, and $\lambda_3$ are the balancing weights.

## 4 EXPERIMENT

### 4.1 Datasets

We mainly conduct experiments on the following seven datasets: (1) **CelebA-HQ** [21] contains $30,000$ real face images, of which $24,000$ are selected for training and the remaining $6,000$ are used for testing. (2) **Metfaces** [17] and (3) **Meticulous** consists of $1,336$ artistic facial avatars and $931$ Chinese ink paintings, respectively, which are randomly divided at a ratio of $8:2$ for training and testing. (4) **Ukiyo-e** [32] contains high-quality Ukiyo-e faces. We randomly select $3,000$ for training and $1,000$ for testing [13]. (5) **AAHQ** [23] consists of facial avatars. We randomly choose $1,500$ samples for training and $1,000$ for testing following [13]. (6) **DeepFashion** [25] consists of $80,000$ fashion images, which are trained and tested according to the official settings. (7) **ADE-20k** [46] contains $20,210$ training images and $2,000$ testing images, which are associated with a 150-class segmentation mask. It is a challenging dataset for most existing methods due to its large diversity.

### 4.2 Implementation Details

**Experimental Settings.** All the experiments are deployed on NVIDIA RTX A6000 GPUs and we use 4 GPUs for training and 1 GPU for inference. In all translation tasks, we specify the size of the input and output images as $256 \times 256$. We build and initialize the visual encoders and correspondence network following [13]. The relevant settings of our diffusion scheduler refer to [9, 45], with diffusion steps $T = 1000$, a linear noise schedule, and a U-Net noise

**Table 1: Quantitative results on the CelebA-HQ, Metfaces, Meticulous, Ukiyo-e, AAHQ, DeepFashion, and ADE-2Ok datasets, where FID and SWD are the main metrics for evaluating the perceptual quality of generated images, supplemented by Texture, Color and Semantic metrics.**

| | CelebA-HQ | | | | | Metfaces | | Meticulous | | Ukiyo-e | | AAHQ | | DeepFashion | |
|---|---|---|---|---|---|---|---|---|---|---|---|---|---|---|---|
| | FID ↓ | SWD ↓ | Texture ↑ | Color ↑ | Semantic↑ | FID ↓ | SWD ↓ | FID ↓ | SWD ↓ | FID ↓ | SWD ↓ | FID ↓ | SWD ↓ | FID ↓ | SWD ↓ |
| SPADE [30] | 31.5 | 26.9 | 0.927 | 0.955 | 0.922 | 45.6 | 26.9 | / | / | 45.6 | 26.9 | 79.4 | 32.1 | 36.2 | 27.8 |
| CoCosNet [43] | 14.3 | 15.2 | 0.958 | 0.977 | 0.949 | 25.6 | 24.3 | / | / | 38.3 | 13.9 | 62.6 | 21.9 | 14.4 | 17.2 |
| CoCosNet-v2 [47] | 13.2 | 14.0 | 0.954 | 0.975 | 0.948 | 23.3 | 22.4 | 34.5 | 24.7 | 32.1 | 11.0 | 62.4 | 22.8 | 13.0 | 16.7 |
| MCL-Net [42] | 12.8 | 14.2 | 0.951 | 0.976 | 0.953 | 23.8 | 24.5 | / | / | 32.4 | 12.4 | 64.4 | 22.2 | 12.9 | 16.2 |
| DynaST [24] | 12.0 | 12.4 | 0.959 | 0.978 | 0.952 | 29.2 | 28.6 | 30.1 | 23.4 | 38.9 | 14.2 | 67.2 | 24.0 | 8.4 | 11.8 |
| Midms [36] | 15.6 | 12.3 | 0.962 | 0.982 | 0.915 | 28.3 | 23.0 | 32.9 | 23.6 | 31.8 | 13.4 | 59.4 | 21.6 | 10.8 | 10.1 |
| MATEBIT [13] | 11.5 | 13.2 | 0.966 | 0.986 | 0.949 | 26.0 | 19.1 | 30.3 | 21.8 | 30.3 | 11.5 | 56.0 | 19.5 | 8.2 | 10.0 |
| PROMOTE (ours) | 11.0 | 12.0 | 0.973 | 0.982 | 0.967 | 22.8 | 18.4 | 29.2 | 21.0 | 28.7 | 11.1 | 54.5 | 18.4 | 7.9 | 9.6 |

estimator. For the proposed global-local contrastive learning, we set the curvature $c = 0.2$ to project the data from Euclidian space to hyperbolic space. The weights $\gamma$, $\lambda_1$, $\lambda_2$, and $\lambda_3$ are set to 0.2, 0.5, 10.0, and 1.0, respectively. We set the batch size to 16 to train our model on DeepFashion for 200 epochs and on the remaining datasets for 100 epochs using the Adam optimizer with the learning rate $1e - 4$.

**Metrics.** To evaluate the translation results comprehensively, we adopt several metrics: (1) *Fréchet Inception Distance* (FID) [35] and *Sliced Wasserstein Distance* (SWD) [16] to evaluate the perceptual quality of generated images. (2) *color, texture, and semantic* cosine similarities based on VGG-19 [38] to evaluate style relevance and semantic consistency of generated images [43].

## 4.3 Comparison with State-of-the-art

We compare the proposed PROMOTE with several SOTA methods, including CoCosNet [43], CoCosNet-v2 [47], DynaST [24], Midms [36] and MATEBIT [13] across seven datasets. All approaches are replicated by adhering to the settings described in their respective papers and source codes.

**Quantitative Analysis.** The quantitative comparison results are illustrated in Table 1. Compared to existing methods, our proposed PROMOTE achieves the best FID and SWD scores on almost all datasets, indicating that the perceptual quality of images generated by our method is superior and closest to real images. This superiority stems from our robust feature extraction encoders and the self-supervised training framework integrating prior information. Furthermore, our method also achieves the best performance on *semantic* and *texture* metrics tested on CelebA-HQ, with improvements of 0.007 and 0.014, respectively. This demonstrates that the structural prior and self-supervised guidance significantly benefits our model in preserving semantics and achieving more consistent styles. Comparatively, since the structural prior (conditional images) contains little color information, the improvement in *color* metric is not as pronounced. The comparison results on the ADE-20k dataset are presented in Table 2, where our method also shows significant gains in challenging scenes with higher diversity and complexity, improving FID and SWD by 0.6 and 0.5, respectively.

**Qualitative Analysis.** Fig. 4 showcases translated images generated by SOTA methods alongside their corresponding input images. The images produced by previous methods exhibit significant flaws, such as geometric distortions, unnatural textures, loss of style, and

**Table 2: Quantitative results on the ADE-20k dataset, where FID and SWD are the main metrics for evaluating the perceptual quality of generated images, supplemented by Texture, Color, and Semantic metrics.**

| | ADE-20k | | | | |
|---|---|---|---|---|---|
| | FID ↓ | SWD ↓ | Texture ↑ | Color ↑ | Semantic↑ |
| CoCosNet | 26.4 | 10.5 | 0.941 | 0.962 | 0.862 |
| CoCosNet-v2 | 25.2 | 9.9 | 0.948 | 0.970 | 0.877 |
| MCL-Net | 24.8 | 9.9 | 0.951 | 0.966 | 0.881 |
| DynaST | 24.6 | 10.1 | 0.960 | 0.967 | 0.875 |
| MATEBIT | 24.3 | 9.7 | 0.957 | 0.973 | 0.880 |
| PROMOTE | 23.7 | 9.2 | 0.966 | 0.978 | 0.893 |

semantic inconsistencies, marked respectively by blue, red, green, and yellow boxes. Benefiting from the global-local contrastive learning and the prior-guided diffusion model, the appearance and detailed style of images translated by our PROMOTE are closest to the exemplars and most semantically consistent with the conditional images, achieving the highest fidelity. Moreover, as illustrated in Fig. 5, thanks to the structural prior of conditional images mathematically modifying the diffusion target and transitions, PROMOTE can generate local attributes (*e.g.* glasses and earrings) with appropriate style even if they are not provided in exemplar images. This capability significantly enhances the practicality of our model.

## 4.4 Ablation Study

This experiment validates the effectiveness of the proposed global-local contrastive learning (*Glo.* for the global part and *Loc.* for the local part) and the prior-guided diffusion model (*Pri.*). Table 3 illustrates the comparison results, where we remove all the components of *Glo.*, *Loc.* and *Pri.* as our baseline model. From the results, we can draw the following conclusions: First, the introduction of *Glo.* decreases FID by 0.3 and 0.6 compared to the baseline model on the CelebA-HQ and Metfaces datasets, respectively. This proves the effectiveness of global cross-domain alignment between exemplars and conditional images in yielding robust domain-invariant features that facilitate accurate cross-domain correspondence. Second, the implementation of *Loc.* alleviates the domain gap at the patch level, resulting in 0.2 and 0.3 FID improvements on the two datasets, respectively. This confirms that *Loc.* aids in better extracting domain-invariant features by optimizing the shortest path

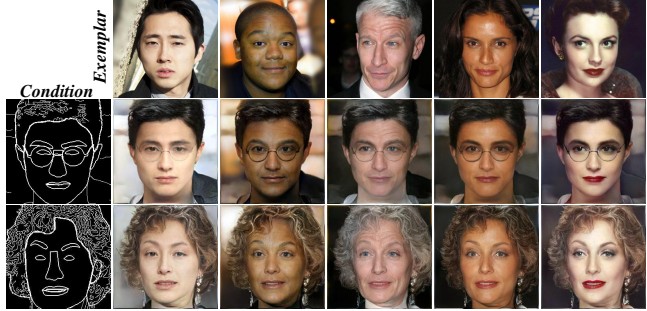

**Figure 4: Qualitative results on the CelebA-HQ, Metfaces, Meticulous, Ukiyo-e, AAHQ, DeepFashion, ADE-20k datasets. These cases demonstrate some of the shortcomings of previous methods: geometric distortions, unnatural texture details, loss of style, and semantic inconsistency, indicated respectively by blue, red, green, and yellow boxes. Notably, our PROMOTE achieves the highest visual fidelity.**

**Figure 5: Generated results for attributes not provided in exemplar. Our PROMOTE can generate local attributes (*e.g.* glasses and earrings) with appropriate style.**

between the semantic correlation distributions of the two images. Third, benefiting from the guidance of target structure prior in the diffusion model, the proposed *Pri.* achieves 0.2 FID improvement on each dataset. This enhancement underscores the value of incorporating structural prior into the diffusion process. Finally, by integrating all three components, our approach reduces the average FID and SWD by 0.75 and 0.7, respectively, which significantly surpasses the baseline model.

### 4.5 Analysis of Global-Local Contrastive Loss

This experiment analyzes the impact of different balancing weight values in Eq. (8) and distance measures on global-local contrastive learning. We vary the values of $\gamma$ to observe the performance changes on CelebA-HQ and Metfaces datasets as presented in Table 4. When $\gamma$ is set to 0, *i.e.* ignoring the optimization by the local

**Table 3: Evaluation results of ablation studies on CelebA-HQ and Metfaces datasets.**

| Dataset | Different Settings | | | Metric | |
|---|---|---|---|---|---|
| | *Glo.* | *Loc.* | *Pri.* | FID ↓ | SWD ↓ |
| CelebA-HQ | - | - | - | 11.8 | 12.6 |
| | ✓ | - | - | 11.5 | 12.4 |
| | - | ✓ | - | 11.6 | 12.4 |
| | - | - | ✓ | 11.6 | 12.3 |
| | ✓ | ✓ | - | 11.3 | 12.2 |
| | ✓ | ✓ | ✓ | **11.0** | **12.0** |
| Metfaces | - | - | - | 22.5 | 19.2 |
| | ✓ | - | - | 21.9 | 18.8 |
| | - | ✓ | - | 22.2 | 18.7 |
| | - | - | ✓ | 22.3 | 19.1 |
| | ✓ | ✓ | - | 21.9 | 18.6 |
| | ✓ | ✓ | ✓ | **21.8** | **18.4** |

**Table 4: Evaluation results of the global-local contrastive loss with different weights $\gamma$ and distance measurements on CelebA-HQ and Metfaces datasets.**

| Dataset | $\gamma$ | 0 | 0.1 | 0.2 | 0.5 | 1 | *Euc.* |
|---|---|---|---|---|---|---|---|
| CelebA-HQ | FID ↓ | 11.4 | 11.2 | **11.0** | 11.1 | 11.3 | 11.5 |
| | SWD ↓ | 12.3 | 12.2 | **12.0** | 12.1 | 12.2 | 12.4 |
| Metfaces | FID ↓ | 22.3 | 22.0 | **21.8** | 22.1 | 22.4 | 22.5 |
| | SWD ↓ | 18.7 | **18.4** | **18.4** | 18.7 | 18.8 | 19.0 |

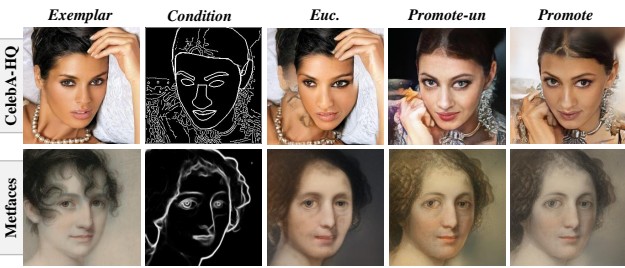

**Figure 6: Comparison of generated images by different settings of our method on CelebA-HQ and Metfaces datasets.**

**Table 5: Evaluation results of the proposed PROMOTE with different training manners on CelebA-HQ and Metfaces.**

| Dataset | Approach | Metric | | Training Speed |
|---|---|---|---|---|
| | | FID ↓ | SWD ↓ | (hour/epoch) |
| CelebA-HQ | PROMOTE-un | 11.7 | 12.9 | 20.347 |
| | PROMOTE | 11.0 | 12.0 | 0.252 |
| Metfaces | PROMOTE-un | 24.3 | 21.2 | 1.023 |
| | PROMOTE | 21.8 | 18.4 | 0.014 |

contrastive loss, the perceptual quality of the translated images is notably the worst. As $\gamma$ increases from 0 to 0.2, performance consistently improves, which confirms the effectiveness of the local contrastive learning since the semantic correlation distributions between two cross-domain images are aligned by minimizing the shortest path measured by Fisher-Rao information. This alignment allows our model to better extract domain-invariant features. However, further increases in $\gamma$ lead to a decline in performance, as an excessively large value of $\gamma$ weakens the global semantic alignment.

Additionally, we perform global-local contrastive learning within the Euclidean distance-based feature space (denoted as *Euc.* in Table 4 and Fig. 6) with optimal hyperparameter settings instead of the hyperbolic representation space. Compared with hyperbolic space, using Euclidean space resulted in an average FID and SWD increase of 0.6 and 0.5 across the two datasets. Moreover, images generated in Euclidean spaces exhibit geometric distortion and unnatural texture details (see Fig. 6). Both qualitative and quantitative results illustrate that performance in Euclidean space is significantly inferior to that in hyperbolic space. This proves that the inaccurate measurement of two cross-domain visual representations by Euclidean distance makes it challenging to explore domain-invariant features, resulting in sub-optimal cross-domain correspondence.

### 4.6 Analysis of Training Manner

This experiment explores two training manners for the diffusion model including unsupervised ("PROMOTE-un") and self-supervised learning. In the unsupervised deployment, necessary perceptual and contextual losses are incorporated into our PROMOTE and optimized through iterative denoising, akin to Midms [36]. The quantitative and qualitative comparison results are presented in Table 5 and Fig. 6, respectively. Benefiting from the explicit guidance of ground truth constructed by self-supervised training, our diffusion model effectively addresses geometric distortion and style/semantic inconsistency, bringing significant performance boosting with 0.7 and 2.5 FID improvements as well as 0.9 and 2.8 SWD improvements on the two datasets, respectively. Moreover, images generated through self-supervised training more faithfully preserve details and styles. Meanwhile, the self-supervised PROMOTE avoids iterative refinement training and eliminates several trivial losses commonly employed in unsupervised deployment for performance enhancement, yielding faster training speed.

## 5 CONCLUSION

This work proposes a novel "**Pr**ior-guided Diffusi**o**n **M**odel with Gl**o**bal-Local Con**t**rastive L**e**arning" (PROMOTE) for exemplar-based image translation. PROMOTE designs global-local contrastive learning to effectively excavate domain-invariant features by aligning cross-domain representations between exemplars and conditional images in hyperbolic space. Furthermore, it emphasizes the target structure as prior information to modify the diffusion transitions and mathematically derives a prior denoising loss. Finally, we tackle this task in a self-supervised manner, constructing deterministic ground truth to train our diffusion model. Our method with theoretical derivations can be generalized to contrastive representation learning and DDPM/DDIM for other generative tasks, facilitating multimedia/multimodal applications. Extensive experiments on seven datasets confirm the effectiveness of PROMOTE, yielding significant performance improvement compared to state-of-the-art methods.

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
