# OpenReview forum: "PROMOTE: Prior-Guided Diffusion Model with Global-Local Contrastive Learning for Exemplar-Based Image Translation"
_acmmm.org/ACMMM/2024/Conference — MM2024 Poster_

### Official Review · Reviewer_W3Fo · 2024-05-23

**Rating:** 5
**Confidence:** 2

**Summary:**

This paper introduces a novel approach called PROMOTE for exemplar-based image translation, which employs a prior-guided diffusion model and global-local contrastive learning to enhance the extraction of domain-invariant features and improve the accuracy of cross-domain correspondences, resulting in state-of-the-art performance across multiple datasets.

**Strengths:**

1. The proposed method significantly outperforms SOTA methods on multiple datasets, delivering superior visual results.
2. The writing is clear and easy to follow, and the topic is both engaging and applicable.
3. Moreover, the method is grounded in solid mathematical theory, ensuring ease of implementation and excellent outcomes.

**Limitations:**

1. Given the strong generative capabilities of existing pretrained diffusion models such as Stable Diffusion, why the authors train a new diffusion model from scratch rather than fine-tuning on the pretrained models? I wonder whether the task can be implemented by just combing IP adapter and controlnet.
2. For reproduction and to facilitate community research, would it be possible for the authors to consider release their code and checkpoints?
3. lack of experimental details, for example, the structure of the utilized modules, such as the conditional encoder and denoising network.
4. This work seems unrelated to multimodal and multimedia.
5. Lack of generalization evaluation: the authors can provide the results with mixing conditions, such as combing edge maps and pose keypoints.

**Suitability:**

2

---

### Official Review · Reviewer_htK8 · 2024-05-24

**Rating:** 5
**Confidence:** 4

**Summary:**

This work proposes a novel "Prior-guided Diffusion Model with Global-Local Contrastive Learning" (PROMOTE) for exemplar-based image translation. PROMOTE designs global-local contrastive learning to effectively excavate domain-invariant features by aligning cross-domain representations between exemplars and conditional images in hyperbolic space. The authors enhance the diffusion model by injecting structural priors and design a novel prior denoising loss to optimize the generation process. Extensive experiments on seven datasets confirm the effectiveness of PROMOTE.

**Strengths:**

1.	The authors reduce the semantic gap between cross-domain images in hyperbolic space, allowing the visual encoders to extract domain-invariant features more effectively.
2.	The prior-guided diffusion model designed by the authors propagates the structural prior to all timesteps in the diffusion process.
3.	The authors designed a novel prior denoising loss to optimize the generation process.
4.	The authors conducted extensive ablation experiments to demonstrate the effectiveness of the method.

**Limitations:**

1. When implementing patch-level contrastive learning, how did the authors divide the patches, and what criteria determine if the division is reasonable?
2. By using data augmentation to implement an unsupervised training strategy, the authors leverage the high similarity between the images before and after augmentation. However, could this approach compromise the model's robustness? For instance, can the model generate desirable results when the input consists of condition images with more complex structures?

**Suitability:**

3

---

### Official Review · Reviewer_GDde · 2024-05-24

**Rating:** 3
**Confidence:** 3

**Summary:**

The paper introduces global-local contrastive learning for the alignment of two cross-domain images. With designed prior-guided diffusion model and prior denoising loss, PROMOTE achieves state-of-the-art performance by constructing reliable guidance in the diffusion model.

**Strengths:**

1. The idea is novel and interesting. In this paper, the authors propose global-local contrastive learning to align images with two domains. And PROMOTE is the first paper that achieves exemplar-based image translation by constructing reliable guidance in the diffusion model.
2. The paper is well-written and easy to follow.

**Limitations:**

For the ablation studies shown in Table. 3, the improvements of FID and SWD are quite small so I am concerned about the effectiveness of the proposed designs.

Although the designs of this paper are novel, the small improvement in the ablation studies cannot show the effectiveness of these designs.

**Suitability:**

3

---

### Meta-Review · Area_Chair_Js3B · 2024-07-01

**Recommendation:** Accept (Poster)
**Confidence:** 5

**Metareview:**

This work has proposed an interesting approach to exemplar-based image restoration with global-local contrastive learning to align two cross-domain images with strong experimental results. The rebuttal has addressed the concerns of the reviewers. In the end, all the reviewers gave "Weak Accept," and the work should be accepted.